# Evaluating the Residual Stress and Its Effect on the Quasi-Static Stress in Polyethylene Pipes

**DOI:** 10.3390/polym14071458

**Published:** 2022-04-03

**Authors:** Na Tan, Liyang Lin, Tao Deng, Yongwu Dong

**Affiliations:** 1School of Aeronautics, Chongqing Jiaotong University, Chongqing 400074, China; jack_linliyang@cqjtu.edu.cn (L.L.); d82t722@cqjtu.edu.cn (T.D.); dongyw234@hotmail.com (Y.D.); 2The Green Aerotechnics Research Institute, Chongqing Jiaotong University, Chongqing 401120, China; 3Department of Mechanical Engineering, University of Alberta, Edmonton, AB T6G 1H9, Canada; 4Chongqing Key Laboratory of Green Aviation Energy and Power, Chongqing 401120, China

**Keywords:** residual stress, mechanical testing, polyethylene, long-term performance

## Abstract

Residual stress is generated during the production process. It can significantly affect the mechanical performance of pressurized polymer pipes. In this paper, six polyethylene (PE) pipes, including three high-density PEs (HDPE) and three medium-density PEs (MDPE) provided by different suppliers, were tested using a one-slit-ring method to measure the residual stress distribution along the hoop direction. Finite element (FE) simulation and mechanical testing were also employed in an iteration process to obtain the mechanical parameters of the six PE pipes. For the same PE pipe code from different suppliers, the results show that the magnitude of the residual hoop stress can be very different, resulting in different mechanical behaviors. In addition, the results are proposed to explain the scenario that was reported previously, i.e., the different critical quasi-static stress (the time-independent stress) levels of the PE pipes with the same pipe code. Since the quasi-static stress is expected to dominate the long-term behavior of the PE pipes, it is of great importance to carefully consider the effect of the residual stress on the determination of the quasi-static stress.

## 1. Introduction

Residual stresses within a pipe are generated as a consequence of a temperature gradient developing during the cooling phase in the extrusion or molding process [1,2,3,4,5]. The outer surface of the pipe is usually quenched in water while the inner surface of the pipe is exposed to static air [6]. Therefore, the outer surface solidifies faster, resulting in a thermal gradient distribution along the pipe wall. The nonuniform solidification process generates residual stress by causing different crystallinity distribution along the pipe wall [6]. Generally, the inner surface is under tensile residual stress while the outer surface is under compressive residual stress [7]. The compressive residual stress within the outer wall of the pipes can benefit their stretching resistance [7]. However, as pointed out by many investigators [8,9,10], the presence of tensile residual stress within the pipes can accelerate the fracture process when conducting a creep rupture test, resulting in the premature failure of the PE pipes. Although the magnitude of the residual stress is not large, it can significantly affect the long-term performance of the pipes, even though the applied stress levels are always very low, typically lower than 5 MPa (hoop direction) [8].

During the last decades, different analytical and computational techniques have been developed to measure the residual stress distribution within pressure pipes. Williams et al. [5] determined the residual stress in a high-density polyethylene (HDPE) pipe by a tube slitting (layer removal) method and evaluated the role of residual stress on fracture. The results were coupled with a semi-elliptical flaw to estimate the stress intensity factor at flaws in the pipe wall. Turnbull et al. [11] compared the residual stress in polycarbonate, filled and unfilled acrylonitrile butadiene styrene (ABS), and nylon using different techniques, including layer removal, hole drilling, and a chemical probe. The comparison showed that layer removal is the most reliable technique, as it provides consistent results, despite it being time-consuming and limited to plate specimens. The slitting method was also widely used in refs. [2,6,7,12,13]. Withers and Bhadeshia [1] have a review on the measuring techniques, including electron diffraction, magnetic and electrical techniques, etc.

In general, residual stresses exist in both hoop and longitudinal directions of pressure pipes. As suggested by Clutton and Williams [14], the presence of residual longitudinal stress in a long pipe can cause an increase in the residual hoop stress. In this study, the short pipes are designed to have a ratio of ring width over thickness of one. For the short pipes with a ratio of ring width over thickness less than two, the effect of the residual longitudinal stress on the residual hoop stress is suggested to be negligible [15]. Based on the methodology introduced by Poduška et al. [2,6], the magnitude and distribution of the residual hoop stress in the PE pipes can be determined by measuring the outer diameter change for one axially slit ring specimen, which was named as a one-slit-ring method for convenience in ref. [7]. The one-slit-ring method is based on the assumption that the distribution of the residual hoop stress on the cross section can be described by an exponential function, which presented to date shows a good fitting to the measured residual hoop stress distribution [2,6].

In general, residual stresses exist in both hoop and longitudinal directions of pressure pipes. As suggested by Clutton and Williams [14], the presence of residual longitudinal stress in a long pipe can cause an increase in the residual hoop stress. In this study, the short pipes are designed to have a ratio of ring width over thickness of one. For the short pipes with a ratio of ring width over thickness less than two, the effect of the residual longitudinal stress on the residual hoop stress is suggested to be negligible [15]. Based on the methodology introduced by Poduška et al. [2,6], the magnitude and distribution of the residual hoop stress in the PE pipes can be determined by measuring the outer diameter change for one axially slit ring specimen, which was named as a one-slit-ring method for convenience in ref. [7]. The one-slit-ring method is based on the assumption that the distribution of the residual hoop stress on the cross section can be described by an exponential function, which presented to date shows a good fitting to the measured residual hoop stress distribution [2,6].

In this study, the residual hoop stress in six polyethylene (PE) pipes of different material characteristics was determined using the one-slit-ring method. We note that the materials used here are the same as those used in a previous study [16], as this is a part of an ongoing study to use a short-term test to quantify the long-term behaviors of the PE pipes. First, this study briefly introduced the one-slit-ring method and then summarized the test results. The test results were used to explain the difference in the measured critical quasi-static stresses for the deformation transitions. The latter was reported previously [16] for the same six PE pipes.

## 2. Experimental

### 2.1. Materials and Sample Preparation

The materials used in the study are six PE pipes of different mass densities and molecular weight distribution. That is, they are either medium- or high-density (MD or HD), and have either bimodal or unimodal molecular weight distribution; hereafter, they denoted as u-MDPE, b-MDPE, u-HDPE, and b-HDPE, in which “u” stands for uni-modal molecular weight distribution and “b”, for bi-modal. In addition, pipes #1 to #4 were obtained from manufacturer A, and pipes #5 and #6 were obtained from manufacturer B. All the pipes have a ratio of outer diameter to wall thickness (SDR) of 11. The above characteristics are summarized in Table 1.

The PE pipes were cut and machined to ring specimens with a width of 6 mm, which is close to the nominal wall thickness, as shown in Figure 1. Three duplicate tests were conducted to ensure the repeatability of the test results.

### 2.2. Monotonic Tensile Test

D-split tensile tests were conducted using a universal test machine (Quasar 100), with the test program and data acquisition controlled by a personal computer. Figure 2 depicts the schematic setup of the D-split test. The tests were under displacement control at a crosshead speed of 30 mm/min. Such a crosshead speed was chosen by taking into consideration the avoidance of heat generation. Two home-made extensometers were mounted on each specimen before the test, in order to record the changes of gauge thickness and width during the test.

In each monotonic tensile test, raw data of the load, stroke, and extensometer readings were recorded as functions of time, based on which engineering stress (*σ*_eng_) was calculated using the following expression:(1)$$\sigma_{\mathrm{eng}}=\frac{F}{T_{0}\times W_{0}}$$
where *F* is the applied load, *W* is the ligament length, and *T* is the ligament thickness. Subscript “0” stands for the initial values of the parameters.

### 2.3. Residual Hoop Stress Measurement

Measurement of the residual hoop stress was carried out by cutting off an arc segment of 120° from each ring specimen using a razor blade, with the guide of a 3D-printed mold shown in Figure 3a. The cutting caused a reduction in the radius of the remaining ring specimen, due to the release of the residual hoop stress. However, due to the viscous nature of the PE pipes, the change in diameter is a function of time [7]. Thus, the diameter was measured 20 min after the cutting to ensure measurement consistency. The diameter change was measured by a non-contact optical comparator (MITUTOYO PH-3500), which provides a measurement resolution of 1 μm. Three duplicates were measured to ensure the repeatability of the test results. The setup for the diameter measurement is shown in Figure 3b.

As mentioned in the Introduction, the one-slit-ring method is based on the assumption that the distribution of the residual hoop stress on the cross section can be described by an exponential function:(2)$$\sigma_{\mathrm{res}}=C_{1}+C_{2}e^{3.2x}$$
where *σ*_res_ is the residual hoop stress, *C*_1_ and *C*_2_ are two constants, and *x* is the normalized position along the ring specimen wall thickness from the inner surface (*x* = 0) to the outer surface (*x* = 1). In order to solve the two constants, *C*_1_ and *C*_2_, it is assumed that the total normal force along the pipe cross section should be zero:(3)$$\int_{0}^{1}\sigma_{\mathrm{res}}Adx=\int_{0}^{1}\left(C_{1}+C_{2}e^{3.2x}\right)Adx=0$$
where *A* is the area of cross section of the ring specimen. With Equation (3) and the conservation of *A* under the small deformation, *C*_1_ can be expressed as a function of *C*_2_:(4)$$C_{1}=-7.354C_{2}$$

In order to solve the value for *C*_2_, another assumption, pure bending under the small deformation, is made according to the curved beam theory [17]. Under the small deformation, the arc length along the neutral plane remains constant:(5)$$R\theta =R^{\prime }\theta^{\prime }$$
where *R*′ and *R* are the radii of the arc along the neutral plane before and after the deformation, and *θ*′ and *θ* are their corresponding central angles, respectively. The neutral plane is subjected to neither compression nor tension, i.e.:(6)$$C_{1}+C_{2}e^{3.2x_{n}}=0$$
where *x_n_* is the normalized position of the neutral plane along the cross section. Thus, *x_n_* can be calculated by substituting Equation (4) into Equation (6) and is equal to 0.6235. Considering an arbitrary arc, with the radii *r* and *r*′ before and after the deformation, along the pipe wall located at a distance y to the neutral axis, the deformation (Δ) can be expressed as:(7)$$\Delta =r\theta -r^{\prime }\theta^{\prime }$$

For the small deformation, together with
(8)r=R−y
and
(9)$$r^{\prime }=R^{\prime }-y$$
the normal strain *ε_x_* has the following expression:(10)$$\varepsilon_{x}=\frac{\Delta }{r\theta }=-\frac{\left(\theta -\theta^{\prime }\right)}{\theta }\frac{R-r}{r}$$

According to Hooke’s Law, the normal stress can be calculated:(11)$$\sigma_{x}=E\varepsilon_{x}=-E\frac{\left(\theta -\theta^{\prime }\right)}{\theta }\frac{R-r}{r}$$
where *E* is the elastic modulus determined with the help of the finite element (FE) simulation, to be detailed later. In addition, the total bending moment (*M_t_*) is equal to the sum of the moments resulting from the normal stress about the transverse direction, and the elementary forces acted on the section should be zero:(12)$$M_{t}=\int -y\sigma_{x}dA=\int \frac{E\left(\theta -\theta^{\prime }\right)}{\theta }\frac{{\left(R-r\right)}^{2}}{r}dA$$
(13)$$\int \sigma_{x}dA=\int -\frac{E\left(\theta -\theta^{\prime }\right)}{\theta }\frac{R-r}{r}dA=0$$

Solving Equations (12) and (13), the total bending moment is represented as:(14)$$M_{t}=\frac{\left(R^{\prime }-R\right)\left(\bar{r}-R\right)EA}{R^{\prime }}$$
with
(15)$$R=\frac{A}{\int \frac{dA}{r}}$$
and
(16)$$\bar{r}=\frac{1}{A}\int rdA$$
where $\bar{r}$ is the centroid. As the ring specimens used in this study have a rectangular cross section, for the small deformation, the centroid is the center of the cross section, and the neutral axis can be calculated by:(17)$$\bar{r}=\frac{R_{out}+R_{in}}{2}$$
and
(18)$$R=\frac{R_{out}-R_{in}}{ln\frac{R_{out}}{R_{in}}}$$
where *R_in_* and *R_out_* represent the radii of the pipe inner surface and outer surface, respectively. The value of *R*′ can be obtained by substituting the *R_in_* and *R_out_* after the deformation into Equation (18). It should be also noted that the change in the outer diameter due to the release of residual stress, Δ*D*, can be approximated as twice the change in the radius of the neutral axis (Δ*D* = 2(*R* − *R*′)) for the small deformation. Alternatively, Equation (14) can be rewritten as:(19)$$M_{t}=\frac{\left(\Delta D/2\right)EA\left(\bar{r}-R\right)}{R-\Delta D/2}$$

On the other hand, the bending moment resulting from the residual hoop stress can be written as:(20)$$M_{r}=\int_{0}^{1}\left(x_{n}-x\right)\sigma_{\mathrm{res}}Ahdx$$
where *h* is the pipe wall thickness. With the assumption that *M_t_* = *M_r_*, the constant *C*_2_ can be calculated as:(21)$$C_{2}=\frac{-\left(D_{0}-h-2R\right)\left(R-R^{\prime }\right)E}{3.382hR^{\prime }}$$

In Equation (21), *D*_0_ and *h* are measured values, while *R* and *R*’ can be calculated based on Equation (18). The constitutive equation for the stress–strain relationship before yielding, as described in [18,19], can be expressed as:(22)$$\sigma =\frac{3}{2\left(1+\nu \right)}E\varepsilon$$
where *σ* and *ε* are the equivalent stress and equivalent strain, respectively, and *ν* is the Poisson’s ratio.

### 2.4. Finite Element (FE) Simulation

The elastic modulus for each PE pipe was determined using FE simulation based on the monotonic tensile test results. The finite element (FE) simulation was conducted using ABAQUS 6.13-4 and one FE model was used for each PE pipe considered in the experimental testing. Taking advantage of symmetry, one-eighth of a ring specimen with the geometry identical to that for the specimen used in the experimental testing and one-quarter of the D-block modeled as an analytical rigid body were coupled for each 3D FE model, as shown in Figure 4. The contact condition was introduced between the inner surface of the ring specimen and the outer surface of the D-block. In addition to the symmetry boundary conditions, displacement control was applied to the block in the FE models to generate deformation. With a fixed Poisson’s ratio of 0.4, FE simulation was used in an iterative process (with an initial guess of 500 MPa) to determine the elastic modulus. The values for the final elastic modulus were chosen so that a satisfactory curve fitting result could be achieved between the engineering stress–stroke and the area change–stroke curves from the mechanical testing and the corresponding results from the FE model.

## 3. Results and Discussion

Figure 5 shows the engineering stress as a function of stroke for all six PE pipes, where the red solid triangle represents the peak stress. In an engineering stress–strain curve, the yield stress for necking samples is the peak stress, while for non-necking samples, it is determined on the curve at which the maximum value of the curvature takes place [20]. The six PE pipes all showed a clear necking process. In other words, the yield point is located where the peak stress occurs. As the figure suggests, the HDPE pipes exhibit both a higher yield stress and modulus than the MDPE pipes. For the PE pipes with the same density, the bimodal PE pipes present a higher yield stress than the unimodal ones. It is worth mentioning that pipes #1 and #5, both u-MDPE but from different manufacturers, show very similar engineering stress–stroke curves; however, pipes #4 and #6, both b-HDPE, show slightly different curves. The yield stress of the PE #4 pipe is noticeably higher than that of the PE #6 pipe, though with similar stiffness in the linear part of the curves. The results also show that all the HDPE pipes have a higher mechanical strength than their MDPE counterparts, consistent with the well-known concept that the strength of PE is mainly governed by the degree of crystallinity [21]. Around the yield point, a load drop occurred in the engineering stress–stroke curve, as the increase in stress was not rapid enough to compensate the narrowing in the specimen’s cross section according to Ref. [20]. This load drop is confirmed by the shape of the curves in Figure 5, which all clearly shows a stress softening after the yield point.

In general, semi-crystalline polymers go through a limited linear deformation before yielding because of their viscous nature. Based on the one-slit-ring method adopted in this study, the elastic modulus is the only unknown. As mentioned in Section 2.4, the values for the elastic modulus were determined using an iteration process in the FE simulation. In other words, the elastic modulus was determined by adjusting the elastic modulus for the FE model; thus, the results generated from the FE simulation matched the linear part of the curves for the engineering stress and the area change versus stroke, in the stroke range below 3 mm. Figure 6 gives an example of the curve fitting process to determine the elastic modulus for pipe #1. The open symbols are from the FE model and the solid lines are from the experimental testing. In the primary axis, the curve of the solid line for engineering stress versus stroke is the same curve as the yellow curve in Figure 5. Within the elastic deformation, the governing parameters are the elastic modulus and Poisson’s ratio. With a fixed Poisson’s ratio and an initial guess of 500 MPa for the elastic modulus, the data from the FE simulation were subtracted to match the engineering stress–stroke (primary axis) and the area change–stroke (secondary axis) curves from the mechanical testing. The values for the final elastic modulus in the FE simulation are chosen so that a satisfactory curve fitting result can be achieved. Table 2 summarizes the elastic modulus of all six PE pipes, obtained by applying the above curve fitting process to their mechanical testing and FE simulation results. Work in ref. [22] has suggested that the tensile modulus of an HDPE pipe material usually ranges between 689 MPa and 896 MPa. Meidani et al. [23] found that the elastic modulus of an MDPE pipe material is about 550 MPa. The values of the elastic modulus in literature are in a similar range of magnitudes as those listed in Table 2. As expected, the elastic modulus of the HDPE pipes is higher than that of the MDPE pipes.

Combined with the elastic modulus and calculated *C*_1_ and *C*_2_ based on Equations (4) and (21), the distribution of the residual hoop stress along the pipe wall thickness was obtained. Figure 7a summarizes the outer diameter changes after cutting off an arc section of 120° from each specimen. The error bars were calculated based on the results of the three duplicates. Figure 7b sets the PE #5 pipe as an example to show the reproducibility of the results on the three duplicate tests. The coincidence of the three curves in Figure 7b indicates good reproducibility of the test results. Hereafter, we chose the average of the three duplicate tests for the analysis.

Figure 8 depicts the distribution of the residual hoop stress for the six pipes. It shows that the maximum compressive residual hoop stress occurs at the outer surface, and the maximum tensile residual stress at the inner surface. The magnitude of the maximum compressive residual stress is about twice that of the maximum tensile residual stress. The values shown in Figure 8 are also close to those reported in Refs. [2,5,9] for the PE pipes. For the PE pipes provided by manufacturer A (pipes #1 to #4), the residual hoop stress in the HDPE pipes is always larger than that in the MDPE pipes (shown from the insert in Figure 8), which is believed to be partly due to the higher elastic moduli of these HDPE pipes. In addition, for pipes with the same density, the pipes from manufacturer B (pipes #5 and #6) have a residual hoop stress of a larger magnitude than the pipes from manufacturer A (pipes #1–#4). While this phenomenon is observed, it is supposed to explain the scenario reported previously in Ref. [16], which is detailed below.

For semi-crystalline polymers, the crystalline phase is stiffer than the amorphous phase; therefore, the deformation process starts in the amorphous phase and involves the crystalline phase only when the stress level is sufficiently high. A previous study [16] proposed a multi-relaxation test (Refs. [16,24]) to detect the transition from the amorphous-phase-dominant deformation to the involvement of the crystalline phase in PE, on the basis of the concept that stress relaxation behavior can be used to reflect the material state of PE under tension. The critical stress for the deformation transition consists of two stress components: one for the time-dependent viscous stress and the other for the time-independent quasi-static stress. The total stress was decomposed using Eyring’s Law [25,26]; the critical quasi-static stresses for the same six 2-inch PE pipes are listed in Table 3.

In Table 3, the critical quasi-static stresses of the six PE pipes for the onset of plastic deformation in the crystalline phase are quite close to their hydrostatic design basis (HDB). Pipes #4 and #6, both b-HDPE but obtained from different manufacturers, are designated to the same HDB with a magnitude of 11.03 MPa. However, the HDB is designated based on a certain range of long-term hydrostatic strength [27]. In other words, the HDB of these two pipes can be the same (11.03 MPa), as long as their long-term hydrostatic strengths are within the same range (10.55–11.93 MPa). Obviously, the effect of higher residual stress in pipe #6 on the designated HDB cannot be assessed directly. As the value of quasi-static stress could play a significant role in the long-term performance of PE, especially for load-carrying applications, the possibility of using the multi-relaxation test to determine the critical quasi-static stress value will greatly benefit the industry in terms of a quick evaluation of the long-term performance of PE. Therefore, we focus on the effect of the higher residual stress on the critical quasi-static stresses obtained from the multi-relaxation test [16,24], which can be used as a means for the preliminary screening or in-service monitoring of pipe performance. A closer examination of Table 3 reveals a lowered critical quasi-static stress of the PE #6 pipe, in comparison with that of the PE #4 pipe. A similar phenomenon was also observed for the PE #1 and #5 pipes. Although those pipes have the same pipe code, their quasi-static stress levels can be very different. Consequently, we inferred that the lowered critical quasi-static stress of the PE #6 and #5 pipes is due to the relative higher tensile residual stress existing in the inner surface of pipes. Meanwhile, many investigators also pointed out [8,9,10] that the presence of tensile residual stress within a pipe can accelerate the fracture process when conducting a creep rupture test, resulting in premature failure. Therefore, it is of great importance to control the residual stress during the manufacturing process. We note that the PE #3 pipe has a bimodal molecular weight distribution and has the same pipe code as that of the PE #1 and #5 pipes; its quasi-static stress is identical to that of the PE #5 pipe, but lower than that of the PE #1 pipe. On the other hand, the residual stress of the PE #3 pipe is close to that of the PE #1 pipe; however, it is lower than that of the PE #5 pipe. The lowered critical quasi-static stress of the PE #3 pipe is still unclear and needs further investigation. 

## 4. Conclusions

The magnitude and distribution of the residual hoop stress within the PE pipes are determined using a one-slit-ring method introduced by Poduška et al. [7]. The results were achieved by measuring the outer diameter change after a slitting process to release the residual stress and determine the elastic moduli in the hoop direction, with the help of experimental testing and FE simulation. The elastic moduli are applied to an exponential function to determine the residual hoop stress.

The mechanical testing results suggest that the HDPE pipes have a better mechanical performance in strength than that of the MDPE pipes. All the PE pipes clearly show a stress softening process, followed by necking, which introduces inhomogeneous deformation. As expected, the FE simulation results suggest that the HDPE pipes are stiffer than the MDPE pipes. In general, the residual hoop stress of the maximum magnitude is compressive stress and occurs at the outer surface. For the PE pipes provided by manufacturer A (pipes #1 to #4), the residual hoop stress in the HDPE pipes is always larger than that in the MDPE pipes, which can be partly interpreted by the higher elastic moduli of the HDPE pipes. In addition, for pipes of the same density category, the PE pipes obtained from manufacturer B (pipes #5 and #6) always have a larger magnitude of residual hoop stress. This phenomenon is supposed to explain the scenario reported previously [16], i.e., the lowered critical quasi-static stress of pipes #5 and #6 is due to the relative higher tensile residual stress existing in the inner surface of the pipes.

## Figures and Tables

**Figure 1 polymers-14-01458-f001:**
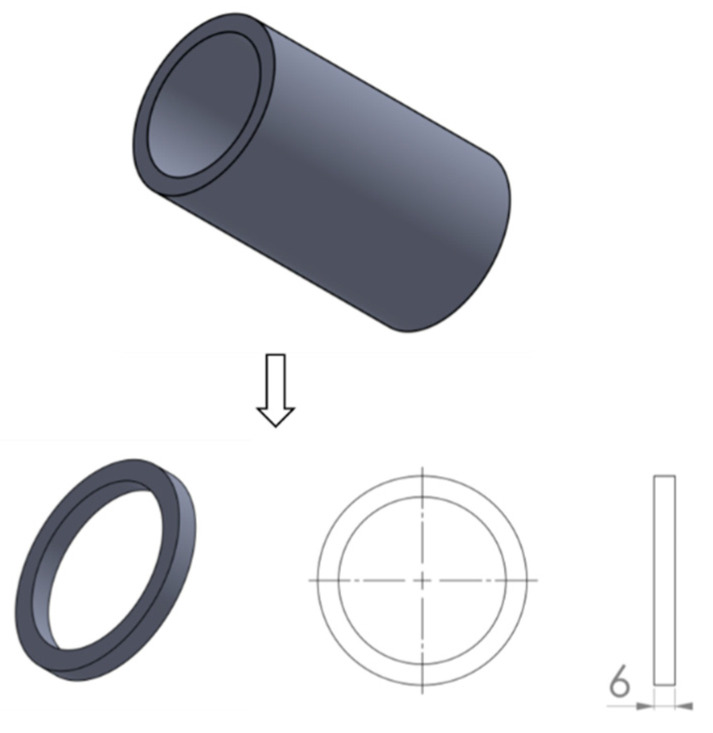
The schematic depiction of the specimen preparation.

**Figure 2 polymers-14-01458-f002:**
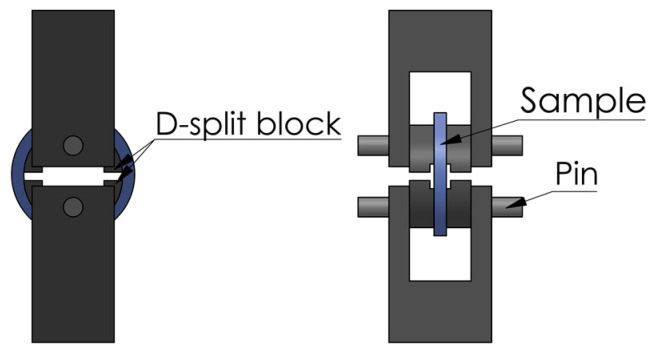
Setup of the D-split tensile test.

**Figure 3 polymers-14-01458-f003:**
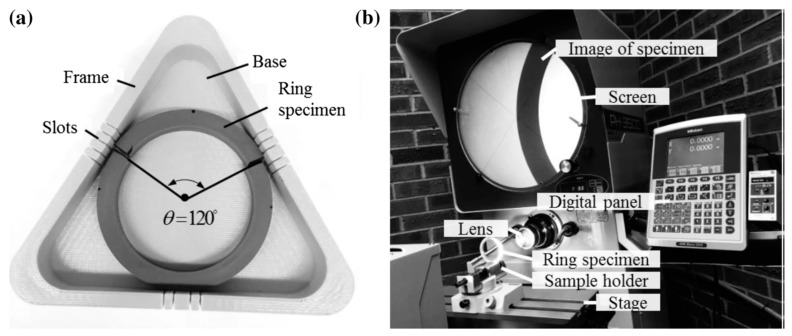
Measurement of the outer diameter change due to the ring slitting: (**a**) a 3D-printed mold for the slitting; and (**b**) the setup of the optical comparator for the measurement of the diameter change. (After ref. [7], with kind permission from Wiley.)

**Figure 4 polymers-14-01458-f004:**
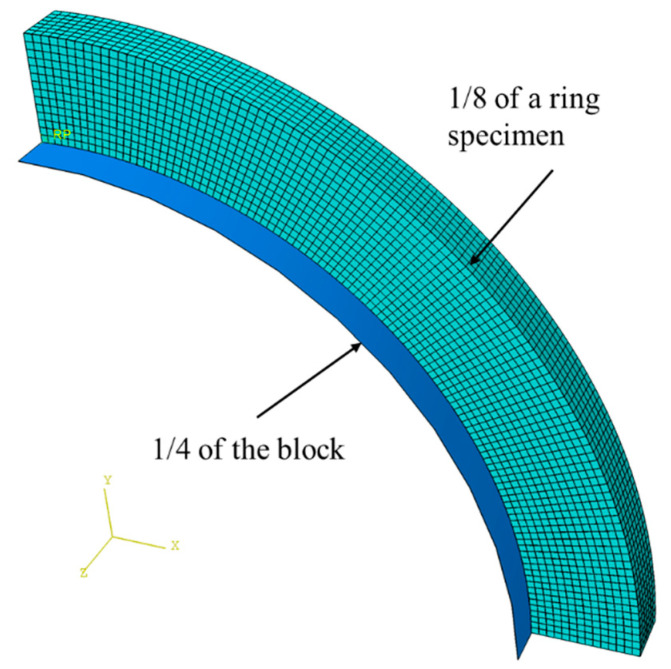
FE model with the mesh pattern.

**Figure 5 polymers-14-01458-f005:**
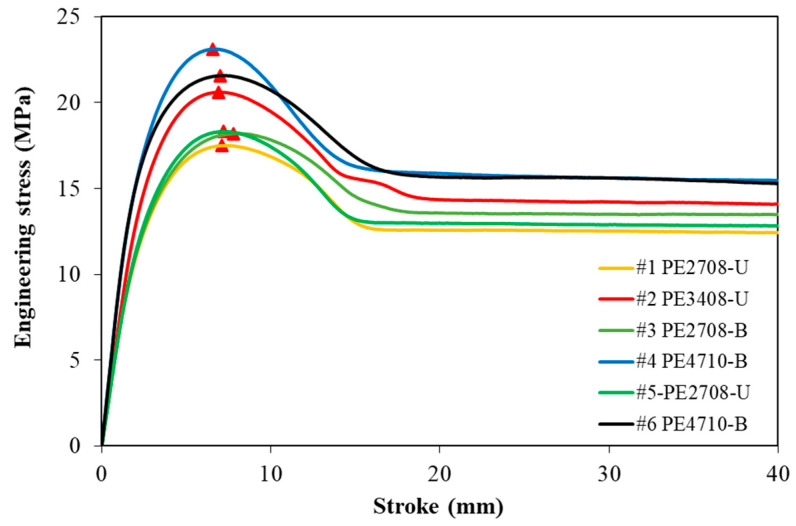
The engineering stress–stroke curves of the six PE pipes. (The red solid triangles mark the peak stress.)

**Figure 6 polymers-14-01458-f006:**
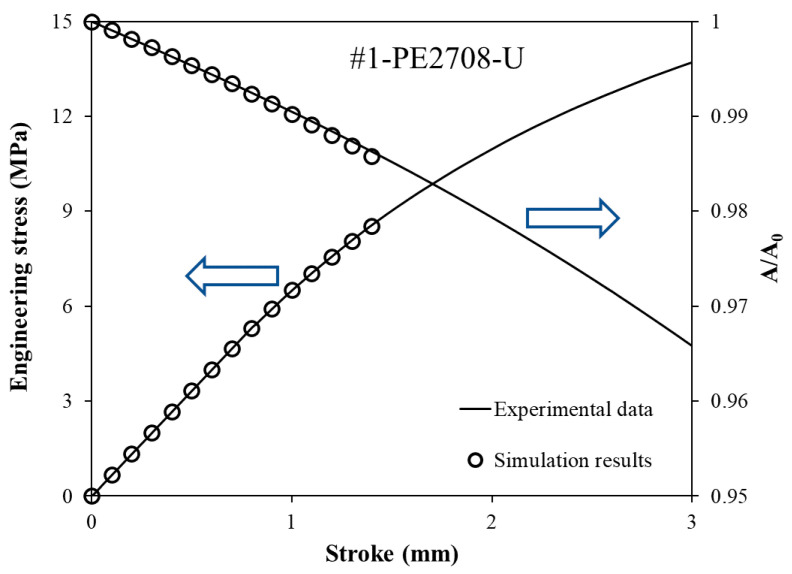
Simulation of the elastic modulus: an example of curve fitting for the PE #1 pipe.

**Figure 7 polymers-14-01458-f007:**
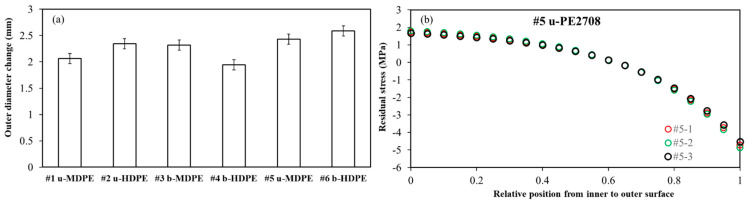
(**a**) Summary of the outer diameter changes after the cutting process and (**b**) the PE #5 pipe as an example of the reproducibility of the results on the three duplicates.

**Figure 8 polymers-14-01458-f008:**
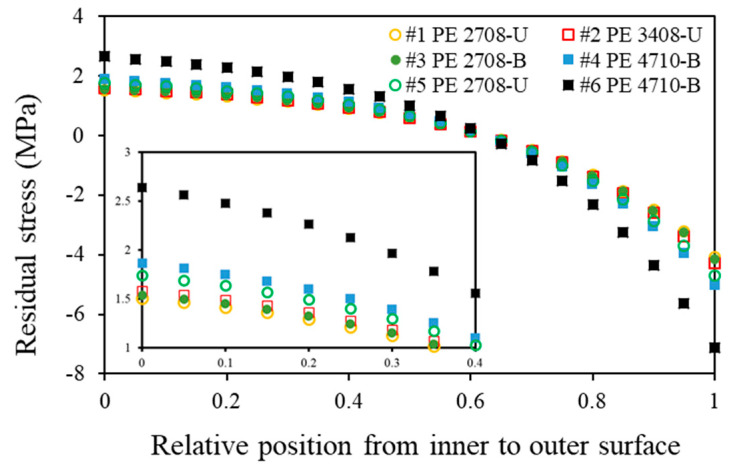
Residual stress distribution along the wall thickness for the six PE pipes.

**Table 1 polymers-14-01458-t001:** Material characteristics of the PE pipes used in the study, which are identical to those in ref. [16].

Material	Pipe Code	Density (g/cc)	Resin Yield Strength (MPa)	Hydrostatic Design Basis (MPa) at 23 °C	Melt Index (g/10 min) at 190 °C/2.16 kg
#1 u-MDPE	PE2708	0.940	19.3	8.62	0.2
#2 u-HDPE	PE3408	0.944 *	22.8 *	11.03	0.08
#3 b-MDPE	PE2708	0.940	19.3	8.62	>0.15
#4 b-HDPE	PE4710	0.949	24.8	11.03	0.08
#5 u-MDPE	PE2708	0.940	19.3	8.62	0.2
#6 b-HDPE	PE4710	0.949	>24.1	11.03	0.08

* Based on the data for PE3608.

**Table 2 polymers-14-01458-t002:** Elastic modulus for the six PE pipe specimens.

Material	Pipe Code	Elastic Modulus (MPa)
#1 u-MDPE	PE2708	570
#2 u-HDPE	PE3408	600
#3 b-MDPE	PE2708	560
#4 b-HDPE	PE4710	795
#5 u-MDPE	PE2708	560
#6 b-HDPE	PE4710	795

**Table 3 polymers-14-01458-t003:** The time-independent quasi-static stress from Ref. [16] and the hydrostatic design basis (HDB) based on the long-term hydrostatic strength for the six PE pipes [27].

Material	Quasi-Static Stress at DB Transition (MPa)	Hydrostatic Design Basis at 23 °C (MPa)	Long-Term Hydrostatic Strength (MPa)
#1 u-MDPE	8.18	8.62	8.27 to 10.55
#2 u-HDPE	10.25	11.03	10.55 to 11.93
#3 b-MDPE	7.25	8.62	8.27 to 10.55
#4 b-HDPE	10.86	11.03	10.55 to 11.93
#5 u-MDPE	7.25	8.62	8.27 to 10.55
#6 b-HDPE	9.88	11.03	10.55 to 11.93

## Data Availability

Not applicable.

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
