# Peer review of "Evaluating the Residual Stress and Its Effect on the Quasi-Static Stress in Polyethylene Pipes"

_polymers, 2022, doi:10.3390/polym14071458_

Round 1

Reviewer 1 Report

The authors have considered my comments. This paper could be accepted for publication The content of the paper improved a lot

Author Response

 Thank you for the supportive comments. We have carefully gone through the whole manuscripts and make corrections when needed. Thank you very much for the excellent comments and suggestions. We hope that the revised manuscript is acceptable for publication in Polymers.

Reviewer 2 Report

The author resubmitted a revised manuscript entitled "Evaluating the residual stress and its effect on the quasi-static stress in polyethylene pipes".

The authors have improved some explanations of methods and results according to reviewers' comments. However, I still disagree with the conclusion of this paper...

The authors try to demonstrate that the difference in the quasi-static stress of PE pipes is caused by the difference in the residual stress in PE pipes. However, I disagree with their conclusions because #5-pipe and #3-pipe, which have almost the same quasi-static stress, showed different values of the residual stress. The authors mentioned in the response letter that #5- and #3-pipe cannot be compared due to the different molecular weight distribution, but if the residual stress is the origin of the different quasi-static stress of PE pipe, all PE pipes should show the same relationship of the quasi-static stress and the residual stress. The authors' assumption can be only applied to #6- and #4-pipes. The different values of both quasi-static stress and residual stress should be due to the morphology distribution in PE pipes. Many researchers have demonstrated that the mechanical properties of semi-crystalline polymers are dominated by morphological parameters! In this manuscript, the residual stress of #6- and #4-pipe were obviously different although the raw PE sample was the same for these pipes, which clearly suggest that the processing condition of these PE pipes were different and the crystallinity (and molecular orientation) distribution of these PE pipes were different. In summary, if the authors discuss the relationship of the residual stress and the quasi-static stress of PE pipes, I believe that the evaluation of morphology such as the crystallinity and the molecular orientation should be required.

Author Response

Dear Reviewer,

Thank you very much for the comments on our manuscript. We have revised the manuscripts accordingly, and highlighted the revised portion using ‘track changes’ function in the manuscript. Below is a summary of our response to your comments.

Response: Thanks for the comments. The residual stress is the one of the origins of the different quasi-static stresses in PE pipes. Other parameters, as mentioned, morphological parameters, can also be the origin of the different quasi-static stresses. The PE #5 and #3 pipes were not compared in the manuscript due to not only their difference in the molecular weight distribution, but also the different manufacturers. We intend to compare the PE pipes with one variable. However, your comments are very valuable. The assumption used for PE #4 and #6 pipes (both bimodal PE4710, but from different suppliers) can also be applied to PE #5 and #1 pipes (both unimodal PE2708, but from different suppliers). But we did not make it very clear in the manuscript. We have revised it accordingly.

For PE #3 pipe, the quasi-static stress is identical to that of PE #5 pipe, but lower than that of PE #1 pipe. On the other hand, the residual stress of PE #3 pipe is close to that of PE #1 pipe, but lower than that of PE #5 pipe. PE #3 and PE #1 pipes are from Manufacturer A, and PE #5 pipe is from Manufacturer B. Therefore, it is possible that manufacturer B introduces higher residual stress in the manufacturing process. Actually, the underlying factors of same quasi-static stress between PE #5 and #3 pipes can be the manufacturing process and molecular weight distribution and so on. Therefore, as the reviewer suggested, further investigation is needed to explore the underlying factor. However, it is out of the research scope, and this is an ongoing study. We still focused on the investigation of using the short-term test to predict the long-term behaviour and we may not wish to draw any conclusions on the results of PE #5 and #3 pipes at this stage. The corresponding revision is as follows and can be found in the manuscript using ‘track changes’ function: A similar phenomenon was also observed for PE #1 and #5 pipes. Although those pipes have the same pipe code, their quasi-static stress levels can be very different. Consequently, we inferred that the lowered critical quasi-static stress of PE #6 and #5 pipes is due to the relative higher tensile residual stress existing in the inner surface of pipes. Meanwhile, many investigators also pointed out [8-10] that the presence of the tensile residual stress within the pipe can accelerate the fracture process when conducting creep rupture test, resulting in the premature failure. Therefore, it is of great importance to control the residual stress during the manufacturing process. Note that PE #3 pipe has a bimodal molecular weight distribution and has the same pipe code as that of PE #1 and #5 pipes, its quasi-static stress is identical to that of PE #5 pipe, but lower than that of PE #1 pipe. On the other hand, the residual stress of PE #3 pipe is close to that of PE #1 pipe, but lower than that of PE #5 pipe. The lowered critical quasi-static stress of PE #3 pipe is still unclear and needs further investigation.

In addition to the above changes, we carefully gone through the whole manuscripts and make corrections when needed. Thank you very much for the excellent comments and suggestions. We hope that the revised manuscript is acceptable for publication in Polymers.

Reviewer 3 Report

I believe that the manuscript has been greatly improved and
now requires publication

Author Response

(The authors gave the same response as above.)

Round 2

Reviewer 2 Report

In the revised manuscript, the authors have added some explanations about the effect of the quasi-static stress on the residual stress. However, the reviewer feels that the relationship between the quasi-static stress and the residual stress of the present pipes cannot be revealed in this paper because the effects of the morphology and primary structure (molecular weight and its distribution) were not examined, as the authors mentioned. The lack of the evaluation of the morphology and primary structure of each pipe is the origin of the uncleared discussion of the relationship between the quasi-static stress and the residual stress in this manuscript. Because the effects of morphology and primary structure on the mechanical properties of semi-crystalline polymers have been studied and the importance of these effects is widely known, the reviewer believes that the evaluation of the morphology and the primary structure is required to be published. However, the reviewer entrusts the decision of the accept or reject to the Editor.

This manuscript is a resubmission of an earlier submission. The following is a list of the peer review reports and author responses from that submission.

Round 1

Reviewer 1 Report

The authors have performed a large amount of research. The results of the work are of practical and scientific interest. I recommend the work for publication 

Reviewer 2 Report

In this paper, the authors discussed the residual stress in polyethylene pipes by combining the FEM and tensile tests. The authors proposed that the magnitude of the residual stress is strongly related to the critical quasi-static stress. 
In general, the difference in the residual stress of the PE pipes should be caused by the morphology distributions such as crystallinity and molecular orientation, as the authors mentioned in the Introduction. However, this paper lacked the morphological evaluation at all. 
The authors concluded that the higher residual stress of #6-pipe than #4-pipe is due to the lower values of the quasi-static stress for #6-pipe than #4-pipe. However, #5-pipe showed higher residual stress than #3-pipe while the quasi-static stress and hydrostatic design basis were the same for both pipes. In addition, the explanation of the details of the quasi-static stress and the hydrostatic design basis is not enough.
I do not recommend this manuscript for publication in Polymers.

Detailed Comments
1. Is the density of each pipe the density of the pipe or the density of the raw material?
2. The color of each pipe in Figure 5 and Figure 8 should be the same.
3. The title of Table 3 is not appropriate.
4. The author should add references to Table 3.

Reviewer 3 Report

Dear Editor,

I have read the manuscript entitled: “Evaluating the residual stress and its effect on the quasi-static 2 stress in polyethylene pipes” and I would like to address following suggestions to the authors:

 In Abstract the first sentence is too long, split it in two.

  • Please compare your results (from lines 206-224 and 263-279), with the literature data.
  • Without reference to the literature in the conclusions and its discussion.
  • What the authors mean by FEM. If they do not discuss FEM in the paper why does it appear in the conclusions section?
  • Line 80, Please add reference 16 and/or other literature references here, as the data in Table 1 is in another table in ref 16.

Minor points:

  • In the manuscript there are many unclear situations, so I ask the authors to check spelling and others grammatical errors:

Line 18, that reported before = that was reported before;  Line 45, Acrylo -nitrile =Acrylonitrile ; Line 53, presence = the presence; Line 54, cause increase  = cause an increase; Line 60, as one-slit-ring = as a one-slit-ring ; Line 74, of different density = of different densities; Line 83, with width of 6 mm = with a width of 6 mm; Line 92, by taking consideration = by taking into consideration; Line 119, x the normalized = x the normalized; Line 188, they all showed = all showed; Line 195, with similarity stiffness = with similar stiffness; Line 196, has higher = have higher; Line 201, show a clearly stress = show a clear stress;